# The Neutrophil Secretome as a Crucial Link between Inflammation and Thrombosis

**DOI:** 10.3390/ijms22084170

**Published:** 2021-04-17

**Authors:** María Amparo Blanch-Ruiz, Raquel Ortega-Luna, María Ángeles Martínez-Cuesta, Ángeles Álvarez

**Affiliations:** 1Departamento de Farmacología, Facultad de Medicina, Universidad de Valencia, 46010 Valencia, Spain; Maria.a.blanch@uv.es (M.A.B.-R.); orlura@uv.es (R.O.-L.); 2Centro de Investigación Biomédica en Red Enfermedades Hepáticas y Digestivas (CIBERehd), 46010 Valencia, Spain

**Keywords:** neutrophil, thrombosis, inflammation, secretome, neutrophil extracellular traps, platelets, extracellular vesicles

## Abstract

Cardiovascular diseases are a leading cause of death. Blood–cell interactions and endothelial dysfunction are fundamental in thrombus formation, and so further knowledge of the pathways involved in such cellular crosstalk could lead to new therapeutical approaches. Neutrophils are secretory cells that release well-known soluble inflammatory signaling mediators and other complex cellular structures whose role is not fully understood. Studies have reported that neutrophil extracellular vesicles (EVs) and neutrophil extracellular traps (NETs) contribute to thrombosis. The objective of this review is to study the role of EVs and NETs as key factors in the transition from inflammation to thrombosis. The neutrophil secretome can promote thrombosis due to the presence of different factors in the EVs bilayer that can trigger blood clotting, and to the release of soluble mediators that induce platelet activation or aggregation. On the other hand, one of the main pathways by which NETs induce thrombosis is through the creation of a scaffold to which platelets and other blood cells adhere. In this context, platelet activation has been associated with the induction of NETs release. Hence, the structure and composition of EVs and NETs, as well as the feedback mechanism between the two processes that causes pathological thrombus formation, require exhaustive analysis to clarify their role in thrombosis.

## 1. Introduction

Cardiovascular diseases are the most common non-communicable diseases and one of the main causes of death throughout the world [1]. In fact, the number of global deaths has steadily increased to 18.6 million in 2019, representing over 30% of total deaths [2]. The prevalence of cardiovascular diseases is likely to increase substantially as a result of population growth and aging. Cardiovascular diseases include myocardial infarction, stroke, and pulmonary embolism, and result from the thrombotic occlusion of vessels. Additionally, patients with other pathologies, such as cancer or Alzheimer’s disease, have a high risk of developing thromboembolism [3,4,5]. Advances have been made in the diagnosis, treatment, and prevention of these events, with anticoagulant therapy and antiplatelet drugs being the current standard of care [6]. Non-vitamin K oral anticoagulants have been shown to be as effective and safe as heparins/warfarin [6,7]. Nonetheless, short-term mortality is 15–20% for pulmonary embolism [8,9] and remains high for any venous thromboembolism, even 3 years after the event [10]. Moreover, approximately 30% of patients with venous thromboembolism have a recurrence within 10 years [2]. Indeed, such diseases reduce the quality of life of patients and exert a grave socioeconomic burden, despite continued improvements in disease treatment and management [1,2]. The use of anticoagulants drugs, which decrease the synthesis of coagulation factors or interrupt the coagulation cascade, and antiplatelet drugs, which block cyclooxygenase or purinergic receptors, fails to fully resolve the complexity of the thrombotic event and the risk of vascular events remains high [11,12]. Therefore, to study the precise process and mechanisms of thrombus formation is of vital importance to the development of novel and more effective therapeutical approaches. 

Platelet function plays a crucial role in thrombotic events; however, although platelets can perform some of their functions individually, it is well known that interactions between them and other vascular cells are essential for numerous other functions [13,14,15,16]. In this context, the role of neutrophils in thrombosis has attracted much attention in recent years. While the recruitment of neutrophils within thrombi has been acknowledged for a long time [17,18], their exact mechanistic roles in this process have not been well characterized. Due to the fact that neutrophils are key for innate immunity and inflammation, and also because of their ability to crosstalk with other vascular cells, including endothelial cells and specially platelets, they may constitute the link between inflammation and the triggering of the initial thrombotic process. Additionally, neutrophils are secretory cells that release well-known soluble inflammatory signaling mediators and other complex cellular structures whose role is not fully understood. In this context, several reports have shown that neutrophil-derived extracellular vesicles (EVs) and neutrophil extracellular traps (NETs) may contribute to thrombosis, and further knowledge of the pathways involved in such cellular crosstalk could lead to the development of new therapeutical approaches. Therefore, the objective of the present review is to determine whether neutrophils—through EVs and NETs—are key factors in the transition from inflammation to thrombosis.

## 2. Role of Neutrophils in the Transition from Inflammation to Thrombosis

The processes of inflammation and coagulation are related, as tissue injuries require both an inflammatory immune response against pathogens and efficient blood clotting to stop bleeding. Neutrophils are essential for the innate immune response against local and systemic tissue insults, and are also major cellular mediators that support inflammation–coagulation interactions [13,19,20]. Upon inflammation, multiple chemotactic stimuli (cytokines, chemokines, etc.) are released to promote neutrophil activation, extravasation, and migration towards the infectious foci. 

One of the first steps of the inflammatory process is leukocyte–endothelium interactions [21,22,23]. Upon a proinflammatory stimulus, leukocytes circulating through the blood vessels—specifically neutrophil—reduce their flow velocity and start tethering to endothelial cells in a process induced by the binding of neutrophil surface ligands, such as P-selectin glycoprotein ligand-1(PSGL-1) or E-selectin ligand-1, with P- and E-selectin molecules expressed on activated endothelial cells. This process is followed by their rolling along the endothelium. Rolling neutrophils develop membrane extensions at their rear end (tethers) and front (slings), which stabilize neutrophil rolling. Subsequently, neutrophils firmly adhere to the endothelium in a process mediated by the binding of neutrophil integrins, such as macrophage-1 antigen (Mac-1) (cluster of differentiation (CD)11b/CD18), with endothelial immunoglobulins, such as intercellular adhesion molecule-1 (ICAM-1). Rolling and adhesion may be followed by crawling and transcellular (through endothelial cells) or paracellular (between endothelial cells) transmigration [21,24,25]. Neutrophil activation can also induce the adhesion of platelets to leukocytes, mainly through the interaction of PSGL-1 on leukocytes with P-selectin on platelets [26,27,28,29]. Platelets and endothelial cells can become activated during these neutrophil–platelet interactions [26]. During the inflammatory responses, numerous soluble factors, such as proteinases, leukotriene A4, and vascular endothelial growth factor, are released from neutrophils and subsequently act on endothelium receptors, therefore inducing changes in the endothelium cytoskeleton that result in cell contraction and loss of barrier integrity. This process induces the release by endothelial cells of cytokines, complement, and antibodies, which further promotes neutrophil recruitment and activation [30]. Upon activation, endothelial cells express several adhesion molecules and consequently initiate the adhesion of platelets to the activated and/or dysfunctional endothelium [26,31]. Simultaneously to these processes, platelets can interact with each other once they are activated, inducing platelet aggregation, which can eventually trigger thrombus formation [26]. All these processes per se, or the release of proinflammatory mediators by any of the cell types involved, can induce the activation of other vascular cells, thus promoting a continuous crosstalk between neutrophils, platelets, and endothelial cells and leading to the transition from inflammation to thrombosis (Figure 1).

Among the different molecular pathways present in neutrophils, the inflammasome is a multiprotein signaling platform that controls the inflammatory response and coordinates host defenses [32]. The inflammasome activates caspases-proteases, which cause the maturation of the cytokines interleukin (IL)-1β and IL-18, and induces an inflammatory form of lytic-type programmed cell death known as pyroptosis, which occurs upon intracellular infection as a part of the defense response [33]. Recent evidence has demonstrated that activation of the inflammasome by an infection stimulus leads to tissue factor (TF) release in the form of microvesicles, which triggers systemic coagulation [32]. This mediator is a high-affinity receptor and cofactor for clotting factor (F)VII/VIIa, and TF and FVIIa form a complex that constitutes the primary initiator of blood coagulation [34]. The study in question showed that TF release from pyroptotic leukocytes initiates systemic coagulation and thrombosis in tissues; furthermore, the authors observed that inflammasome activation-induced coagulation requires caspase-1 but does not require IL-1β and IL-18 [32]. This is consistent with previous findings in which caspase-1 activation by adenosine triphosphate (ATP) promoted TF-positive microvesicles’ release [35].

Nuclear factor of the ĸ-chain in B cells (NF-ĸB) is another central mediator of inflammation, and is fundamentally involved in the molecular link between inflammatory and thrombotic processes [36]. NF-ĸB activity is not only triggered by inflammatory cytokines, such as tumor necrosis factor alpha (TNF-α) or IL-1, but also by bacterial cell wall components, such as lipopolysaccharides (LPS), by viruses, and even by physical stress conditions [37,38,39]. This could explain the various types of target genes that are upregulated or induced after neutrophil NF-ĸB activation, including cytokines and chemokines, immune receptors, adhesion molecules, antiapoptotic genes, acute phase proteins, various enzymes, stress response genes, and coagulation regulators. In fact, TF, FVIII, urokinase-type plasminogen activator (uPA), and plasminogen activator inhibitor-1 (PAI-1) are induced by NF-ĸB activation [40,41,42,43]. Thus, NF-ĸB contributes to coagulatory events, not only via cellular activation processes, but also by transcriptional induction of proteins of the plasmatic coagulations cascade. This provides another molecular explanation for the functional molecular link between inflammation and thrombosis that contributes to an increased cardiovascular risk in situations of acute or chronic inflammation [36].

Platelets are cells prepared to respond to stimuli with rich surface membrane receptors, but not for the “de novo” synthesis of mediators, since they are anucleated cells. In contrast, neutrophils are cells with the genetic machinery necessary for the synthesis of new molecules that could be key for understanding the thrombotic processes. Therefore, it is important to analyze what structures are released by neutrophil and how they influence platelets. Neutrophils can be induced to release proinflammatory and procoagulant molecules, such as cytokines, TF, matrix metalloproteinases, damage-associated molecular patterns (DAMPs) - including histones, high-mobility group protein B1 (HMGB1) or DNA fragments - and also EVs and NETs [44,45,46,47,48,49,50,51,52]. These mediators have an influence on several aspects of thrombus formation, including platelet activation and adhesion, as well as activation of the intrinsic and extrinsic coagulation pathways.

## 3. Neutrophil Extracellular Vesicles (EVs)

Neutrophils are secretory cells whose cytosol contains a large variety of granules formed sequentially during myeloid cell differentiation [53,54]. There are three generally recognized types of granules that differ from each other in their content, function, and signals required for secretion: azurophilic granules, specific granules, and gelatinase granules [53,54,55,56]. Nevertheless, these granules are not totally exclusive, as there is some overlapping among them. The content of all these granules is important if we are to understand the role of neutrophils in triggering thrombotic events, since some proteins and mediators released by these granules may activate platelets and induce platelet aggregation and/or coagulation. Azurophilic granules contain a potent mixture of antimicrobial effectors, including neutrophil elastase (NE), myeloperoxidase (MPO), and cathepsin G [53,54]. Specific granules and gelatinase granules enclose proteins, which support migration, and some antimicrobial proteins, such as collagenase, gelatinase, lactoferrin. or uPA [53,54]. The specific content of these granules is detailed elsewhere [54,56,57,58,59], and this field of research is constantly updated as new proteins are discovered in the content of the different granules. Recently, it has been described that, when neutrophils release the content of their granules, they additionally release EVs into the extracellular space. EVs were originally described as “platelet dust” [60], but over the years their role in cellular communication has become patent [61]. Therefore, they are no longer regarded as “cell dust”, but rather as messengers between adjacent and distant cells, since EVs serve as vehicles for the intercellular exchange of biological material and information [62]. In this way, EVs have become an important focus of attention in thrombosis research. These EVs are formed by a phospholipid bilayer, proteins of the cytosol, and of the cellular membrane, nuclear material, and non-coding RNAs [63,64]. The four distinct populations of EVs described are exosomes, microparticles, microvesicles, and apoptotic bodies. EVs can promote neutrophil activation and migration, specific immune responses, inflammatory reactions, atherogenesis, plaque rupture, and thrombosis. The content of EVs varies considerably according to their release activation stimulus and posterior function [53]. Neutrophils generate EVs either spontaneously or in response to various stimulants during immune responses, which bears testimony to the importance of their role in the effector functions of neutrophils. The stimulants for EVs release can be classified in three categories: bacteria and bacterial products (LPS), cytokines and chemokines (IL-8, TNF-α), and exogenous compounds (Phorbol 12-myristate 13-acetate (PMA), zymosan) [65,66]. The functions and content of EVs are more diverse and dynamic than those of other granules, which has led to them being highlighted as crucial elements of the role of neutrophils in thrombus formation. In addition, and because of their heterogeneity, it is important to study the different populations of EVs stimulated by the diverse endogenous stimuli that are present in different patophysiological conditions. For instance, upon bacterial infection during the innate immune response, neutrophils release EVs that contain arachidonic acid (AA), which stimulates platelets, thus supporting the link between proinflammatory and proaggregating mediators [67]. EVs usually contain inflammatory cytokines, ICAM-1, PSGL-1, TF, complement receptor 3 (CR3), metalloproteases, and nucleic acids (e.g., tRNAs, mRNAs, miRNAs) [68,69,70,71,72]. In this regard, many studies have shown non-coding RNAs—for instance, miR-21, miR-126 or miR-146a—as potential therapeutic targets or biomarkers in the progression of cardiovascular diseases [73,74,75].

It is currently known that neutrophils can induce thrombosis through the release of soluble factors from their granules (Figure 2A). Neutrophils are a source of cytokines and release soluble factors into the milleu. Proinflamatory cytokines, such as IL-1, IL-6, TNF-α, interferon (IFN)-α, and IFN-Υ, induce the expression and release of TF. These cytokines not only trigger procoagulant activity but also inhibit the anticoagulant pathway of thrombomodulin/protein C and affect fibrinolysis by upregulating uPA and PAI-1. The role of cytokines in inflammation and thrombosis has been acknowledged [76], while that of EVs is beginning to be clarified. Thus, they can promote thrombosis via various mechanisms (Table 1), including: the presence of phosphatidylserine (PS) in the phospholipid bilayer of EVs from activated neutrophils [77,78,79] promotes platelet activation and formation of blood clots (Figure 2B) [80]; EVs express several integrins (such as Mac-1) that can interact with platelets, inducing platelet P-selectin expression and their activation (Figure 2C) [81]; activation of the coagulation cascade by both intrinsic and extrinsic pathways due to the presence of TF and polyphosphates (PolyP) in the membrane of EVs (Figure 2D) [77,82]; and the presence of MPO on EVs is associated with thrombosis, since MPO causes endothelial damage, promoting the adherence and activation of platelets (Figure 2E) [64] (Table 1).

In the last decade, microvesicles have become increasingly important in the context of thrombosis, with many reports published about them (Table 1). As we have discussed in the previous section, the presence of TF on the surface of microvesicles is widely associated with thrombotic events. Alternatively, other reported evidence suggests that the presence of PolyP in microvesicles promotes thrombosis through a TF-independent route [82]. PolyP is a highly anionic linear polymer synthesized from ATP, and affects numerous steps in the coagulation cascade, including the activation of FXII, thus enhancing the activation of FV and increasing the activity of the thrombin-activated fibrinolysis inhibitor and inhibiting the TF pathway inhibitor. In addition, PolyP is thought to enhance fibrin clot structure stability [83,84,85]. Another study revealed the presence of enzymes that synthetize leukotriene B_4_ (LTB_4_) in neutrophil exosomes, and the presence of the primary substrate AA, suggesting an active exosomal LTB_4_ synthesis that can promote platelet activation [86,87]. A proteomic analysis in microvesicles demonstrated the presence of 116 proteins, 31 of which were significatively different between baseline and two days after thrombosis induction. The authors in question discovered microvesicles with fibrinogen on their surface [88]. Fibrinogen is a six-chain protein precursor of the clot structural protein, fibrin, and dimer of α, β, and γ polypeptides [89,90]. Therefore, they suggested that the presence of fibrinogen pointed to the implication of microvesicles in thrombotic processes by playing an essential role in platelet aggregation, inflammation and wound healing. Another proteomic study [91], in this case performed on exosomes of unstimulated and LPS-stimulated neutrophils, identified 271 unique proteins, the majority of which were identified in both conditions, thus demonstrating the constant composition of exosomes. However, 16 of these proteins were differentially expressed upon LPS neutrophil stimulation: phosphatidylcholine-sterolacyltransferase, tenascin-X, thrombospondin-1 (TSP-1), annexin A7, neurogenic locus notch homolog protein 2, lactotransferrin, integrin-linked protein kinase, fibrinogen A-α chain, serpin peptidase inhibitor clade B member 6, lipocalin 2, α-1-acid glycoprotein 3, complement 3, profilin-1, protein S100A9, triosephosphate isomerase, and β_2_ integrins. Some of these proteins have already been associated with coagulation and thrombotic processes—for instance, thrombospondin-1 [92], fibrinogen A-α chain [89], and complement 3 [15]—but we cannot rule out that the others may also be involved in said processes. Another proteomic study executed on secretomes of a carotid atherosclerotic plaque and a non-atherosclerotic mammary artery identified a total of 162 proteins [93], 25 of which exhibited significant differences in their levels in the secretome, and most of which, including extracellular superoxide dismutase and peroxiredoxin-2, were downregulated. The abovementioned proteins are antioxidant enzymes that can prevent oxidative stress and, thus, endothelial dysfunction [94,95]. Consequently, if these enzymes are downregulated, endothelial dysfunction can occur and platelet activation and adhesion can be induced. In contrast, the levels of other proteins, such as neutrophil defensin 1, apolipoprotein E, clusterin, and zinc-alpha-2-glycoprotein, were increased. Neutrophil defensin 1 induces the binding of fibrinogen and thrombospondin-1 to platelets and causes platelet aggregation [96]. Clusterin participates in lipid transport and is considered an adhesion molecule, and thus induces cell aggregates [97]. Zinc-alpha-2-glycoprotein is an adipokine and has been shown to be involved in endothelial dysfunction processes [98,99,100] (Table 1).

Hence, it is evident that the neutrophil secretome contains a large number of different proteins, which are not only released upon proinflammatory or prothrombotic stimuli, but also under basal or non-pathological conditions. This also signifies that there are many pathways by which thrombi can be induced, and it is likely that not all of them have been described yet. Therefore, further investigations are required to determine the biological mechanisms by which EVs are released, and the exact roles played by their released mediators. In this regard, omics approaches, such as proteomics, metabolomics, and transcriptomics, have enabled the overall characterization of complex global biological systems at the molecular level and their alterations during pathological processes. In this context, proteomics has emerged as a useful tool for analyzing the proteins present in the secretome of neutrophils and those involved in the pathogenesis of thrombosis. Moreover, new biomarkers may have the potential to improve risk stratification, diagnosis, and/or treatment.

## 4. Neutrophil Extracellular Traps (NETs)

Another mechanism by which neutrophils may represent a link between inflammation and thrombosis is the release of NETs [15,102,103]. NETs were first described in 2004 by Brinkman and colleagues as a novel immune defense mechanism of neutrophils, acting as a physical barrier that prevents the spread of pathogens and facilitates the clearance of microbes by phagocytosis and by exposing them to high concentrations of antimicrobial proteins [104]. NETs are a network of extracellular decondensed filaments of DNA, from the nucleus or mitochondria [105], and are associated with histones and cytosolic and granule proteins, such as histones themselves, NE, MPO, calprotectin, cathelicidins, cathepsin G, leukocyte proteinase 3, lactoferrin, gelatinase, lysozyme C, HMGB1, peptidyl arginine deiminase 4 (PAD4), defensins, and actin [51,52,106,107]. From all the studies published on the subject, it can be concluded that the composition of NETs varies depending on the different neutrophil stimulus employed. In this regard, whether and how differences in NETs’ composition impact NET function remains to be investigated. Two different mechanisms of NET release have been described; one called NETosis, which occurs through the cell death process, and another termed non-lytic NETosis, which leads to a more rapid release of NETs in the absence of cell death [108,109]. In short, NETosis consists of neutrophils arresting their actin dynamics and depolarizing, leading to nuclear delobulation and the disassembling of the nuclear envelope, followed by nuclear chromatin decondensing into the cytoplasm and mixing with cytoplasmatic and granule components [108,110]. Subsequently, the plasma membrane permeabilizes and NETs expand into the extracellular space in a process that takes place 3–8 h after neutrophil activation. In contrast, non-lytic NETosis occurs within minutes, is accompanied by the release of granule proteins through degranulation, and does not lead to rupture of the cell membrane [111,112,113].

As shown in Figure 3, the process of NETosis is initiated by a great variety of stimuli, all yet to be determined, and which interact with NETosis-inducing receptors. PMA was the original stimulus by which NETs were discovered [104]; however, a large number of other NETosis stimuli have already been described, including bacteria, fungi, viruses, auto-antibodies or immune complexes, IL-8, TNF-α, miR-146a, hydrogen peroxide, urate and cholesterol crystals, Oxidized low-density lipoprotein (oxLDL), cigarette smoke, ionophores, complement-derived peptides, HMGB1, and activated platelets [74,104,105,108,112,114,115,116,117,118,119,120,121,122,123,124]. These aforementioned stimuli act through their binding or the activation of receptors, such as Toll-like receptor (TLR) 2, 4, 6, 7, and 8; dectin 2 (while dectin 1 inhibits it); receptor for advanced glycation end products (RAGE); CR3; FcγR; CD36; and PSGL-1 [112,115,123,125,126,127,128,129]. Independently of the stimulus, NETs formation does not require transcription [130]. Specifically, PMA directly binds to protein kinase C (PKC), inducing calcium release from intracellular stores. Consequently, the Raf- mitogen-activated protein kinase (MEK)- extracellular signal-regulated kinase (ERK) pathway is activated, thus promoting nicotinamide adenine dinucleotide phosphate (NADPH) oxidase generation of reactive oxygen species (ROS) [108,131,132,133]. Downstream, as part of the protein complex known as the azurosome, MPO promotes the release of NE from azurophilic granules. Subsequently, MPO and NE translocate to the nucleus, where MPO binds to chromatin and synergizes with NE, which partially degrades histones, thus contributing to chromatin decondensation [134]. The involvement of NE in NETosis has also been demonstrated in patients with mutations in the cysteine protease cathepsin C, which processes NE into its mature form, since NETosis proved to be defective in these patients [135,136]. The mobilization of intracellular calcium and ROS generated by NADPH have been described as inducers of PAD4 activation and translocation to the nucleus, where PAD4 contribute to chromatin decondensation through citrullination of the histones, which is a well-acknowledged aspect of NETosis [137,138,139,140]. PAD4 modifies arginine residues of histones H3 and H4 by citrullination, which causes a loss of positive charge and of hydrogen bonds between DNA and histones, thus leading to decondensation and swelling of the nucleus [141,142]. Indeed, PAD4 was originally suggested to be a specific and universal mediator of NET formation; however, some recent findings have suggested that NETs can be released in a PAD4-independent manner [119]. The mechanism leading to rupture of the nuclear and plasma membranes is not yet well understood, but recent studies point to the contribution of the pore-forming gasdermin D [143]. In this sense, after proteolytic cleavage, gasdermin D was shown to enhance the release of neutrophil proteases into the cytosol by forming pores in the granule membranes, allowing the translocation of NE to the nucleus, as well as the release of the content of granules, therefore allowing granules and cytosolic proteins to mix with the NETs scaffold [143,144]. Furthermore, cleaved gasdermin D formed pores in the plasma membrane, facilitating NETs release. Interestingly, it has been suggested that gasdermin D-dependent NET formation is independent of ROS, NE, and PAD4 [143,144]. It has also been shown that the ATP-binding cassette transporter A1/G1 and downstream NLRP3 inflammasome activation led to NETs formation [145]. An additional pivotal event required for NETosis is autophagy [146]. This process occurs independently from ROS production and involves the cessation of mammalian target of rapamycin (mTOR) (Figure 3).

Besides the acknowledged role against pathogens of NETs in the immune system, there is also evidence pointing to NETs as a new link between immunity and thrombosis, since the treatment of mice with DNase I and PAD4 or NE inhibitors prevents thrombus formation in a similar way to heparin [102]. Moreover, patients with acute thrombosis exhibit lower levels of plasma DNase I activity [147]. In this context, the inflammatory microenvironment and the rupture of the plaque induce neutrophils to release NETs expressing bioactive TF, thus causing thrombus formation in patients with myocardial infarction [148]. Some mechanisms that trigger NETs formation leading to myocardial infarction include elevation of the monocyte chemoattractant protein-1, plasma glucose levels, and complement activation [149,150,151].

NETs can be toxic to endothelial cells, as they promote endothelial dysfunction, which in turn activates the endothelium and induces NETs formation, triggering a vicious cycle that results in further damage [152,153,154]. In this regard, an injured endothelium can lead to atherosclerosis characterized by the accumulation of lipoproteins, cholesterol crystals, and oxLDL, molecules that in turn can induce NETosis. This network plays a significant role in the progression towards plaque formation in a process that can obstruct blood flow over time [119]. Interestingly, due to the ability of NETs to act as a scaffold for platelet and erythrocyte adhesion, it can trigger the formation of large aggregates that may obstruct small blood vessels. In the case of erythrocytes, their adhesion to NETs results in a red thrombus [102]. Nevertheless, the crucial role of NETs in the multiple facets of thrombosis is mostly attributed to the interactions of neutrophils and NETs with platelets. NETs can capture platelets, promoting platelet activation and aggregation. NETs have also been shown to promote thrombin formation by providing a scaffold for procoagulant molecules, such as von Willebrand Factor (VWF), fibronectin, fibrinogen, FXII, TF, and pro-coagulant EVs, including TF-EVs [15,102,155,156,157,158,159]. As such, the DNA–histones backbone is thought to add stability to the fibrin scaffold in thrombi [160], despite the fact that the NETs scaffold itself can support clot formation without the presence of fibrin [161]. Furthermore, NETs components promote the gene expression of coagulation factors [162]. NETs seem to promote vessel occlusion in both fibrin-dependent and -independent manners, and also activate the coagulation cascade via intrinsic and extrinsic pathways [15,103]. In addition to their function as a scaffold, many of the components of NETs can induce platelet activation and blood coagulation, thus triggering thrombus formation.

The multiple mechanisms by which NETs can induce thrombus formation have been summarized in Table 2. DNA can activate coagulation by FXII (a protein that promotes coagulation and mobilizes endothelial cell Weibel–Palade bodies that contain VWF, P-selectin, and FXIIa) of the intrinsic pathway and enhance the activity of coagulation serine proteases [163,164]. Furthermore, nucleic acids interfere in the inhibition of clot clearance by impairing fibrinolysis through the inhibition of plasmin-mediated fibrin degradation forming complexes with plasmin and fibrin [165] (Table 2).

Histones induce TF expression in vascular endothelial cells, macrophages, and monocytes, which would activate coagulation via the extrinsic pathway [166,167]. In addition, H3 and H4 specifically trigger platelet activation and aggregation, the release of procoagulant polyphosphates from platelet granules, and an increase of local thrombin generation by interacting with platelets directly via TLR2 and TLR4 [102,154,168,169]. Histones can also bind to VWF, fibrinogen, and fibrin to recruit platelets and erythrocytes [170]. Conversely, histones have also been shown to increase thrombin generation through modulation or degradation of plasma anticoagulants. Indeed, histones interact with thrombomodulin and protein C, thus inhibiting thrombomodulin-mediated protein C activation and further boosting plasma thrombin generation [171] (Table 2).

NE degrades the anticoagulant protein tissue factor pathway inhibitor (TFPI) but also enhances FXa activity and proteolytically activates platelet receptors to increase platelet accumulation [103] (Table 2). Similarly, cathepsin G cleaves to TFPI, thereby impeding clot clearance, and has recently been identified as an inducer of platelet activation via protease-activated receptor 4 (PAR4) [103]. Cathepsin G-induced activation seems to involve the P2Y12 platelet receptor, the integrin glycoprotein (GP)IIbIIIa, and Syk kinase [172]. Interestingly, a recent article supports a regulatory role of NETs in the production and molecular integrity of TSP-1. NE and cathepsin G induce the proteolysis of TSP-1 to a smaller isoform, which exhibits better hemostatic properties than its precursor molecule. NETs enhance the production of TSP-1 and prevent its complete degradation after extracellular exposure to excessive protease concentrations [173] (Table 2). This is important because TSP-1 is a protein released by platelets and endothelial cells, which participates in hemostasis. In this context, PAD4 released by NETs has been reported to citrullinate a disintegrin and metalloproteinase with thrombospondin type-1 motif-13 (ADAMTS13), thus reducing its activity [174]. ADAMTS13 is involved in the clearance of VWF–platelet complexes, contributing to thrombosis resolution [175], so the inhibition of ADAMTS13 activity by PAD4 is an important prothrombotic mechanism. On the other hand, HMGB1 can promote platelet–leukocyte adhesion and NETs release [46,117,123,124,126] (Table 2).

NETs are important to platelet activation and thrombotic processes, but activated platelets are also capable of triggering NET formation [103,112], thus implicating platelet-neutrophil interactions not only in thrombosis, but also in inflammation and related disorders. In turn, in response to various classic platelet agonists, such as adenosine triphosphate (ADP), collagen, AA, and thrombin, activated platelets induce the production of NETs in a platelet number-dependent manner [123,176]. Thus, platelets induce NETosis through pathways involving TLR4, HMGB1, and P-selectin [27,112,117,123,124]. The compelling evidence provided above demonstrates the complex crosstalk between platelets, neutrophils, and NETs release that triggers a vicious circle leading to pathological thrombus formation.

**Table 2 ijms-22-04170-t002:** Relation of NETs components to thrombosis.

NETs Component	Relation to Thrombosis	References
DNA	Coagulation cascade activation	[163,164,165]
	Endothelial Weibel-Palade bodies mobilization	
	Clot clearance inhibition	
Histones (H3, H4)	TF expression	[102,154,166,167,168,169,170,171]
	Recruit platelets and erythrocytes	
	Thrombin generation	
	Platelet activation and aggregation	
MPO	Endothelial dysfuntion	[101]
NE	TFPI degradation (avoid clot clearance)	[103,173]
	Recruit platelet	
	TSP-1 production	
Cathepsin G	TFPI degradation (avoid clot clearance)	[103,172,173]
	Platelet activation	
	TSP-1 production	
PAD4	ADAMTS13 inactivation (avoid clot clearance)	[174,175]
HMGB1	Platelet-leukocyte adhesion	[46,117,123,124,126]
	NETs release	

NETs: Neutrophil extracellular traps. DNA: Desoxyribonucleic acid. TF: tissue factor. MPO: Myeloperoxidase. NE: neutrophil elastase. TFPI: Tissue factor pathway inhibitor. TSP-1: Thrombospondin-1. PAD4: peptidyl arginine deiminase 4. ADAMTS13: A disintegrin and metalloproteinase with thrombospondin type-1 motif-13. HMGB1: high-mobility group protein B1.

Moreover, clinical studies have placed the focus on NETs as a crucial source of biomarkers in plasma and tissue samples from patients with diverse prothrombotic and thrombotic diseases, as a key to preventing and treating these pathologies. The common biomarkers identified in studies that have analyzed NETs release were: DNA (double-strain DNA or cell-free DNA), nucleosomes, citH3 or citH4, MPO, NE, and diverse molecular complexes formed by the combination of these biomarkers (such as nucleosomes, DNA-citH3 complexes, DNA-MPO complexes, nucleosomes-MPO complexes, MPO-NE complexes, NE-DNA complexes, etc.) (Table 3). In fact, the presence of the abovementioned NETs has been described in patients suffering from arterial (acute coronary syndrome, coronary artery disease, and stroke) and venous (deep vein thrombosis, pulmonary embolism) thrombotic episodes, and other related syndromes (thrombocytopenia, septic shock with disseminated intravascular coagulation, Behçet’s disease, Cushing disease) [150,177,178,179,180,181,182,183,184,185,186,187,188,189,190,191,192,193,194,195]. These clinical studies correlated a specific molecular pattern of the NETs with the severity of tissue damage using parameters and signs, such as infarct size, stenosis grade, electrocardiogram disturbances, stroke scores, thrombus stabilization and growth, thrombin potential ratio, tronponin T peak, hypercoagulability markers, protein C reactive, and glucose levels (Table 3). It has also been reported that NETs stimulate fibrin formation and deposition, and that fibrin colocalizes with NETs in blood clots [15,102,103]. Interestingly, the presence of DNA and MPO as NETs forms have also been described in patients with hypertension, considered a common risk factor for cardiovascular disease rather than a thrombotic event, with NETs correlating with the homocysteinemia in these hypertensive patients. In other studies, patients with cancer (such as pancreatic, lung, brain, breast, and ovarian cancers) have a higher risk of developing venous thromboembolism, and experimental data indicate that NETs play an important role in this association [3,196]. Furthermore, NETs have been implicated in cancer progression; they entrap cancer cells in the vasculature, which correlates with increased tumor metastasis [197].

## 5. Conclusions and Future Challenges

Neutrophils are activated upon some inflammatory stimuli, causing complexes, such as the inflammasome and the transcription factor NF-ĸB, to induce the release of a whole series of mediators that mediate both inflammatory and thrombotic processes, including EVs and NETs. By themselves or through their components, these machineries activate platelets to trigger thrombi formation. Furthermore, their composition, and especially that of EVs, can vary significantly depending on the stimulus that activates the neutrophils and promotes their release.

Many previous studies have blocked various molecules or employed knockout strains to inhibit the release of EVs and/or NETs, and some or all of these molecules may represent therapeutic targets. Among the currently identified and promising targets for inhibiting NETosis are ROS production, PAD4, and DNase, currently the most used inhibitor. The administration of DNase1 disrupts NETs and reduces disease severity in some mouse models [198]. Although DNase1 disrupts the DNA-formed scaffold, it does not have an effect on other NETs components that can attach to vessel walls and continue causing damage. Further studies are needed to clarify these issues.

EVs have recently been proposed, not only for therapeutics, but also for drug delivery purposes due to their unique properties, which allow a drug to be directed through a target cell. To generate theragnostic EVs, diagnostic agents and therapeutic molecules can be incorporated into EVs [199]. In addition, EVs and NETs have emerged as alternative biomarkers for the detection of cardiovascular diseases, thus they may provide information regarding the pathology and the efficacy of a treatment regimen. Nevertheless, more research is necessary before EVs and NETs can be used as reliable biomarkers in thrombotic diseases.

Thus, precise studies must be carried out in order to analyze these structures (EVs and NETs), not only in samples from patients with different pathological conditions related to thrombosis, but also in in vitro studies employing neutrophil stimulation with different endogenous mediators. Such approaches may be sure to throw light on the elements of these machineries that trigger disorders, with the goal of developing effective new therapeutical approaches that are specific for each condition.

## Figures and Tables

**Figure 1 ijms-22-04170-f001:**
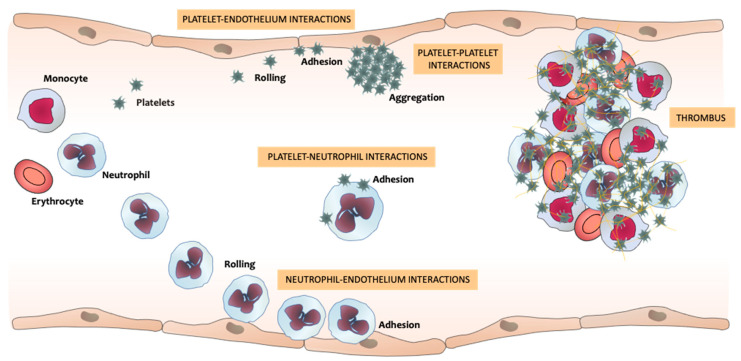
Neutrophils’, platelets’, and endothelial cells’ interactions that promote the transition from inflammation to thrombosis. Upon an inflammatory signal, neutrophils circulating across the vessels reduce their flow velocity and start rolling along endothelial cells, eventually adhering to them. This process can induce the activation and recruitment of further neutrophils, in addition to endothelium activation or dysfunction, which can trigger platelet adhesion to the endothelium and to neutrophils. The interactions between these three types of vascular cells, and with circulating monocytes and erythrocytes, trigger the thrombus formation that can produce vessel occlusion.

**Figure 2 ijms-22-04170-f002:**
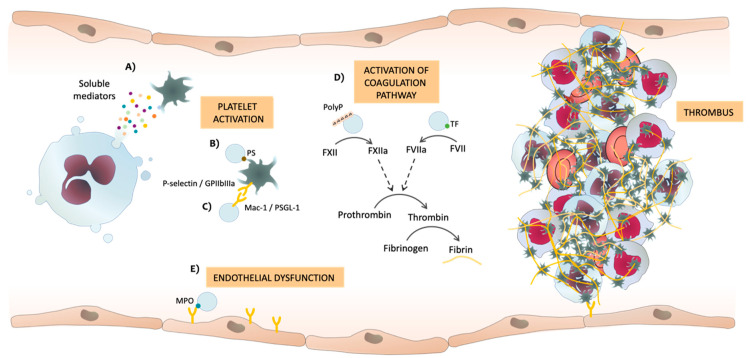
Potential mechanisms by which soluble mediators or neutrophil extracellular vesicles (EVs) released by neutrophils can induce thrombosis. (**A**) Soluble mediators released by neutrophils, (**B**) phosphatidylserine (PS) and (**C**) different adhesion molecules present in the membrane of EVs interact with platelets, thereby inducing platelet activation and aggregation. (**D**) EVs activate both intrinsic and extrinsic coagulation pathways. Polyphosphates (PolyP) and tissue factor (TF) in EVs can activate the coagulation cascade, thus producing thrombin from its inactive form of pro-thrombin, and thrombin finally induces the formation of fibrin from fibrinogen. (**E**) Myeloperoxidase (MPO) in EVs can induce endothelial dysfunction, which triggers platelet activation and aggregation. F: factor; GP: glycoprotein; PSGL-1: P-selectin glycoprotein ligand-1.

**Figure 3 ijms-22-04170-f003:**
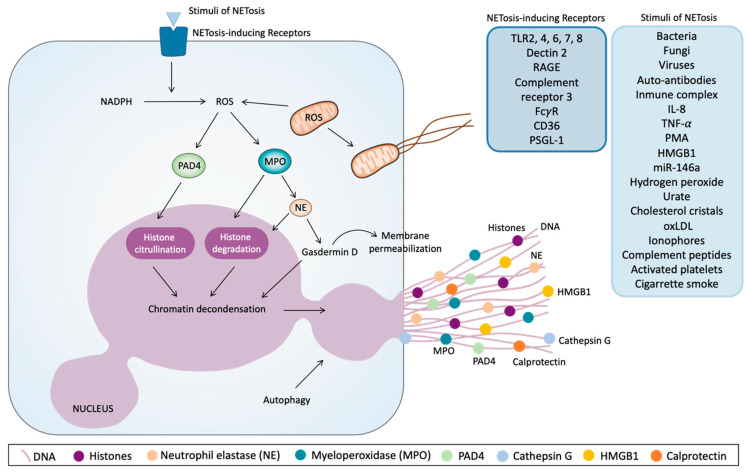
Molecular pathways of the release of neutrophil extracellular traps (NETs). Different stimuli and several receptors can induce NETs’ release. One of the first intracellular events that initiate this phenomenon is reactive oxygen species (ROS) production via nicotinamide adenine dinucleotide phosphate (NADPH) or by mitochondria. ROS induce the activation and translocation to the nucleus of peptidyl arginine deiminase 4 (PAD4), which citrullinates histones and contributes to chromatin decondensation. On the other hand, ROS induce myeloperoxidase (MPO) activation, thus promoting neutrophil elastase (NE) release from granules and their translocation to the nucleus, where NE can degrade histones and promotes chromatin decondensation. NE also activates proteolytically gasdermin D, which can facilitate NE translocation to the nucleus, thus increasing chromatin decondensation. Gasdermin D promotes the disassembly of the nuclear envelope, after which nuclear chromatin decondenses into the cytoplasm, mixing with the cytoplasmatic and granule components and also the permeabilization of plasma membrane, thus allowing NETs to expand into the extracellular space. Autophagy is another cellular mechanism that can trigger the release of NETs. DNA: Deoxyribonucleic acid. TLR: Toll-like receptor. RAGE: Receptor for advanced glycation end products. CD: cluster of differentiation. PSGL-1: P-selectin glycoprotein ligand-1. IL: interleukin. TNF-α: Tumor necrosis factor alpha. PMA: Phorbol 12-myristate 13-acetate. HMGB1: high-mobility group protein B1. miR-146a: micro Ribonucleic acid 146a. oxLDL: oxidized low-density lipoprotein.

**Table 1 ijms-22-04170-t001:** Secretome components that could trigger thrombosis.

Secretome Component	Secretome Compartment	Relation to Thrombosis	References
PS	EVs/NMP	Platelet activation	[77,78,79,80]
Adhesion molecules	NMP	Cell-cell adhesion	[81]
MPO	EVs	Endothelial dysfunction	[101]
TF	NMP	Coagulation cascade activation	[77,81]
PolyP	EVs	Coagulation cascade activation	[82]
LTB_4_	Exosomes	Platelet activation via AA	[86,87]
Fibrinogen	NMP/Exosomes	Platelet aggregation	[88,89,91]
TSP-1	Exosomes	Platelet aggregation	[91,92]
Complement 3	Exosomes	Atherosclerotic lesions	[15,91]
Neutrophil defensin	Secretome	Platelet aggregation	[93,96]
Clusterin	Secretome	Cell-cell adhesion	[93,97]
Zinc-alpha-2-glycoprotein	Secretome	Endothelial dysfunction	[93,98,99,100]

PS: Phosphatidylserine. EVs: Extracellular vesicles. NMP: Neutrophil-derived microparticles. MPO: Myeloperoxidase. TF: Tissue factor. PolyP: Polyphosphates. LTB_4_: Leukotriene B_4_. AA: Arachidonic acid. TSP-1: Thrombospondin-1.

**Table 3 ijms-22-04170-t003:** NETs components detected in patients with different pathologies related to thrombosis.

Biomarker	Patients	Sign Correlated to the Biomarker	References
DNA	Arterial thrombotic events and diseases	Infarct size	[150,177,178,179,180,181,182,183,184,185,186,189,190]
		Stenosis grade	
		Severe stroke scores	
		Hypercoagulability markers	
		High glucose levels	
		Thrombus age	
	Venous thromboembolic events	Thromboembolism extent	[187,188]
		C-reactive protein	
	Hypertension	Homocysteinemia	[193]
	Other diseases	ITP: Platelet CD62 *^†^	[191]
		Behçet’s disease: Vascular involvement	[192]
		Cushing disease: ETP ratio	[194]
Nucleosomes	Arterial thrombotic events and diseases	Infarct size	[178,185,190]
		Stenosis grade	
		Severe stroke scores	
		High glucose levels	
	Venous thromboembolic events	Thromboembolism extent	[187]
	Other diseases	Cushing disease: ETP ratio	[194]
citH3/H4	Arterial thrombotic events and diseases	Infarct size	[180,181,184,189,190]
		Stenosis grade	
		Severe stroke scores ^†^	
		High glucose levels ^†^	
		Thrombus age, stabilization and growth	
	Venous thromboembolic events	Lactate levels	[195]
	Other diseases	ITP: Platelet CD62 *^†^	[191]
MPO	Arterial thrombotic events and diseases	Stenosis grade ^‡^	[185]
	Venous thromboembolic events	C-reactive protein	[195]
		D-dimer	
	Hypertension	Homocysteinemia ^‡^	[193]
	Other diseases	Behçet’s disease: Vascular involvement ^‡^	[192]
		Cushing disease: ETP ratio	[194]
NE	Arterial thrombotic events and diseases	Thrombus stabilization and growth	[184]

NETs: Neutrophil extracellular traps. DNA: Desoxyribonucleic acid. citH3/H4: citrullinated histone H3/H4. MPO: Myeloperoxidase. NE: neutrophil elastase. ITP: Primary immune thrombocytopenia. CD: cluster of differentiation. ETP: Endogenous thrombin potential. * Negative correlation. ^†^ DNA-citH3 complexes. ^‡^ DNA-MPO complexes.

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
