# Peer review of "The Neutrophil Secretome as a Crucial Link between Inflammation and Thrombosis"

_ijms, 2021, doi:10.3390/ijms22084170_

Round 1

Reviewer 1 Report

I really enjoyed reading the manuscript by Maria Amparo Blanch-Ruiz et al. because it provides a nice overview on the topic. The authors provide a well-written manuscript considering both English and clarity of the content. The manuscript generally is well organized and easy to follow, however, in comparison with the rest of the review, the first two sections (the introduction, and the description of the role of neutrophils in the transition from inflammation to thrombosis) need to be enriched in general, it may add more value to the review. In addition, I suggest to add a section of “future challenges” or “clinical implications”, or include a paragraph in the conclusion to suggest how NMV or NETs studies might lead to new therapeutic opportunities for the prevention of thrombotic complications in cardiovascular diseases. I like the effort the authors have put into the tables.

Major comments

  1. Introduction

The introduction of cardiovascular diseases is really short and without any detail, for example it would be nice to add a brief but comprehensive description of epidemiological data, incidence in society, available therapies and their effectiveness. I have the feeling that it is necessary to frame the problem and to emphasize the need to investigate new targets and/or develop new approaches. Moreover, brain blood flow reductions and thrombus formation occur in the vast majority of dementia patients, including those with Alzheimer’s disease, maybe to mention also this aspect (in introduction or in the conclusion) might strength and broaden the importance and implication of the review.

  1. Role of neutrophils in the transition from inflammation to thrombosis
  • Line 60-67: the leukocyte recruitment cascade is not really well detailed and some imperfections need to be revised regarding the adhesive interactions between leukocytes (neutrophils) and the inflamed endothelium. I also suggest to update the literature cited with more recent papers/reviews.
  • Line 69-76: the review would benefit from a short paragraph describing how neutrophils activate the endothelial cells and most importantly, how the activated endothelium contributes to the activation of neutrophils/platelets and the formation of thrombi (mentioned again in line 166-167 or 212-213 or 314-16).
  • Regarding pyroptosis, it would be more precise to add a definition or a brief description.
  • Line 116-119: the list of various pro-inflammatory and pro-coagulant molecules is quite long and maybe two/three examples of the 9 articles cited in this phrase regarding some aspects of thrombus formation in which these molecules are involved could be added, just to give an overview.
  1. Neutrophil exosomes and neutrophil-derived microvesicles (NMVs)
  • Line 147-150: the authors mention the release of microvesicles upon different stimuli as important initiators of the thrombotic process. A more detailed description of these stimuli and neutrophil activation during cardiovascular diseases could help clarify the importance of these neutrophil factors in thrombosis.
  • Line 157: the authors mention the presence of mRNA/miRNA in the microvesicles released from neutrophils, however this issue is not further developed. miR-146a has been associated with cardiovascular complications, and recently, Arroyo A.B and colleagues have described miR-146a as a pivotal regulator of NET formation promoting thrombosis (PMID: 32586906).

Minor comments

  • Line 111: not clear what context the authors mean by “In this context (…)”
  • Line 182: “(…) which many reports published (…)” probably the authors meant with
  • Table 1: typing error in “Atherosclerotic lessions
  • Line 361: “(…) citrullinates (…)
  • Figure 2: “Potential mechanism (…)” The authors provide more than one possible mechanism.
  • Figure 2: there is the number 70 beneath the thrombus drawing. Not clear to what it is referred to.

Author Response

Referee 1

“I really enjoyed reading the manuscript by Maria Amparo Blanch-Ruiz et al. because it provides a nice overview on the topic. The authors provide a well-written manuscript considering both English and clarity of the content. The manuscript generally is well organized and easy to follow, however, in comparison with the rest of the review, the first two sections (the introduction, and the description of the role of neutrophils in the transition from inflammation to thrombosis) need to be enriched in general, it may add more value to the review. In addition, I suggest to add a section of “future challenges” or “clinical implications”, or include a paragraph in the conclusion to suggest how NMV or NETs studies might lead to new therapeutic opportunities for the prevention of thrombotic complications in cardiovascular diseases. I like the effort the authors have put into the tables”.

We very much appreciate that referee 1 considers that he/she has really enjoyed reading our manuscript because it provides a nice overview on the topic, that we provide a well-written manuscript considering both English and clarity on the content and that he/she likes the effort that we have put into the tables.  Care has been taken to modify the new version in accordance with his/her suggestions and to discuss the subjects he/she raises more extensively. According to his/her suggestion, we have included a paragraph in the conclusion to suggest how NMV or NETs studies might lead to new therapeutic opportunities for the prevention of thrombotic complications in cardiovascular diseases (“future challenges”) (Page 14, lines 451-464). We believe the comments have improved the manuscript.

Major Comments:

Point 1. ”Introduction” The introduction of cardiovascular diseases is really short and without any detail, for example it would be nice to add a brief but comprehensive description of epidemiological data, incidence in society, available therapies and their effectiveness. I have the feeling that it is necessary to frame the problem and to emphasize the need to investigate new targets and/or develop new approaches. Moreover, brain blood flow reductions and thrombus formation occur in the vast majority of dementia patients, including those with Alzheimer’s disease, maybe to mention also this aspect (in introduction or in the conclusion) might strength and broaden the importance and implication of the review.

Response 1. According to referee’s suggestions, these aspects have been incorporated in the introduction (Pages 1-2, lines 33-49).

Point 2. Role of neutrophils in the transition from inflammation to thrombosis

Line 60-67: the leukocyte recruitment cascade is not really well detailed, and some imperfections need to be revised regarding the adhesive interactions between leukocytes (neutrophils) and the inflamed endothelium. I also suggest updating the literature cited with more recent papers/reviews.

Response 2. The leukocyte recruitment cascade and the literature cited have been updated (Pages 2, line 78-88).

Point 3. Line 69-76: the review would benefit from a short paragraph describing how neutrophils activate the endothelial cells and most importantly, how the activated endothelium contributes to the activation of neutrophils/platelets and the formation of thrombi (mentioned again in line 166-167 or 212-213 or 314-16).

Response 3. The short paragraph suggested by the referee has been included (Pages 2-3, line 88-95).

Point 4. Regarding pyroptosis, it would be more precise to add a definition or a brief description.

Response 4. A brief description of pyroptosis has been inserted (Page 3, lines 113-114).

Point 5. Line 116-119: the list of various pro-inflammatory and pro-coagulant molecules is quite long and maybe two/three examples of the 9 articles cited in this phrase regarding some aspects of thrombus formation in which these molecules are involved could be added, just to give an overview.

Response 5. Several molecules have been removed. The specific explanation by which some of these molecules (neutrophil elastase, histones, HMGB1 and DNA) induce thrombus formation was already included on the section 4 entitled “neutrophil extracellular trap” (Page 10, lines 374-399).

Point 6. Neutrophil exosomes and neutrophil-derived microvesicles (NMVs)

Line 147-150: the authors mention the release of microvesicles upon different stimuli as important initiators of the thrombotic process. A more detailed description of these stimuli and neutrophil activation during cardiovascular diseases could help clarify the importance of these neutrophil factors in thrombosis.

Response 6. The stimuli that induce the release of microvesicles (now named extracellular vesicles due to referee number 2 suggestion) have been included (Page 4, line 172-177).

Point 7. Line 157: the authors mention the presence of mRNA/miRNA in the microvesicles released from neutrophils, however this issue is not further developed. miR-146a has been associated with cardiovascular complications, and recently, Arroyo A.B and colleagues have described miR-146a as a pivotal regulator of NET formation promoting thrombosis (PMID: 32586906).

Response 7. We thank the reviewer by his/her suggestion to further develop the concept of the presence of mRNA/miRNA in the microvesicles released by neutrophils. The issue has been completed and miR-146a and miR-21 and miR-126 have been incorporated as pivotal regulators of NET formation promoting thrombosis (Page 5, lines 185-187; page 8, line 293; page 9, line 326, figure 3 under stimuli of NETosis).

Minor Comments:

Point 8. Line 111: not clear what context the authors mean by “In this context (…)”

Response 8. The sentence has been modified and “in this context” has been removed (Page 4, line 137).

Point 9. Line 182: “(…) which many reports published (…)” probably the authors meant with

Response 9. The point of the referee is valid and the word “which” has been replaced by “with” (Page 5, line 212).

Point 10. Table 1: typing error in “Atherosclerotic lessions”

Response 10. “Lessions” has been changed to “lesions” (Page 7, table 1).

Point 11. Line 361: “(…) citrullinates (…)”

Response 11. “Citrullinates” has been corrected to “citrullinate” (Page 10, line 394).

Point 12. Figure 2: “Potential mechanism (…)” The authors provide more than one possible mechanism.

Response 12. We agree with the referee, and the word “mechanism” has been modified to “mechanisms” (Page 5, line 203).

Point 13. Figure 2: there is the number 70 beneath the thrombus drawing. Not clear to what it is referred to.

Response 13. The new version of the manuscript that includes the suggestions of the referees does not contain the number 70 beneath the thrombus drawing (Page 5, line 202, Figure 2).

Reviewer 2 Report

The authors present a well written review on the role of the neutrophil secretome as a crucial link between inflammation and thrombosis. The review is a timely article with the renewed interest in neutrophils as drivers of inflammation and thrombosis. However, I have a number of concerns with the article in its current form. Some terminologies used throughout the review and in the figure legends are confusing, with the review lacking details on the role of other neutrophil secreted factors (cytokines) in thrombosis.

  1. The authors do not discuss in great detail non vesicle or NET associated factors secreted by neutrophils such as cytokines and their role in thrombosis. Many of these factors influence EV and NET release from neutrophils. The title "secretome" is not fully reflective by the content of this current version of the review.
  2. The article discusses neutrophil derived microvesicles using the terminology of NMv. I suggest that the author clarify further throughout the manuscript what is meant by this. Are these NMV’s extracellular vesicles? If so its recommended that the term neutrophil derived EV is used as microvesicles are a subfraction of EVs. Also, the term exosome is used. Exosomes, microvesicles and apoptotic bodies are all collectively EVs with their own unique protein cargo and cellular functions. It’s important that this is communicated within the review as currently its confusing.
  3. Table 1 should be updated to clarify if the studies investigated EVs, MVs, exosomes or microparticles.
  4. While the review primarily focuses on thrombosis in general, there is no mention of the role of NETs and neutrophil derived EVs as potential drivers of cancer associated thrombosis (CAT). Cancer is associated with an increased risk in thrombosis, particularly VTE, with chemotherapy, surgery and cancer histology all increasing the risk. Neutrophils are a central player in this, and the review would benefit from this inclusion.

Author Response

Referee 2

The authors present a well written review on the role of the neutrophil secretome as a crucial link between inflammation and thrombosis. The review is a timely article with the renewed interest in neutrophils as drivers of inflammation and thrombosis. However, I have a number of concerns with the article in its current form. Some terminologies used throughout the review and in the figure legends are confusing, with the review lacking details on the role of other neutrophil secreted factors (cytokines) in thrombosis”.

We fully appreciate the comments of reviewer 2 and thank her/him for considering that “we present a well written review on the role of the neutrophil secretome as a crucial link between inflammation and thrombosis and that the review is a timely article with the renewed interest in neutrophils as drivers of inflammation and thrombosis”. The manuscript has been modified according to the referee’s suggestions, which we believe have helped to improve our work. The specific responses to the comments of the referee are as follows:

Point 1. The authors do not discuss in great detail non vesicle or NET associated factors secreted by neutrophils such as cytokines and their role in thrombosis. Many of these factors influence EV and NET release from neutrophils. The title "secretome" is not fully reflective by the content of this current version of the review.

Response 1. Soluble factors such as cytokines and their role in thrombosis have been incorporated into the text (Page 5, lines 189-194).

Point 2. The article discusses neutrophil derived microvesicles using the terminology of NMv. I suggest that the author clarify further throughout the manuscript what is meant by this. Are these NMV’s extracellular vesicles? If so its recommended that the term neutrophil derived EV is used as microvesicles are a subfraction of EVs. Also, the term exosome is used. Exosomes, microvesicles and apoptotic bodies are all collectively EVs with their own unique protein cargo and cellular functions. It’s important that this is communicated within the review as currently its confusing.

Response 2. We accept the point of the reviewer and, therefore, the text has been modified throughout the manuscript by replacing the term “neutrophil derived microvesicles” by “neutrophil derived extracellular vesicles (EVs)” (Page 1, lines 17, 18, 20, 24; page 2, line 64, 67; page 4, line 148, 166, 167, 169, 170, 171, 173, 174, 175, 177; page 65 line 180, 182, 184, 196, 197, 200, 201, 203, 205, 206, 207, 209; page 7, line 253, 260 and page 14, line 448, 450, 466).

Point 3. Table 1 should be updated to clarify if the studies investigated EVs, MVs, exosomes or microparticles.

Response 3. Table 1 has been updated to clarify whether each study investigated EVs, MVs, exosomes or microparticles (Page 7, lines 252-255, table 1).

Point 4. While the review primarily focuses on thrombosis in general, there is no mention of the role of NETs and neutrophil derived EVs as potential drivers of cancer associated thrombosis (CAT). Cancer is associated with an increased risk in thrombosis, particularly VTE, with chemotherapy, surgery and cancer histology all increasing the risk. Neutrophils are a central player in this, and the review would benefit from this inclusion.

Response 4. Following referee’s suggestion, the relationship between cancer and thrombosis has been included in the introduction (Page 1, line 37-38) and the role of NETs as potential drivers of CAT is described in neutrophil extracellular trap section (Page 12, lines 433-437).

Reviewer 3 Report

In this manuscript (ID: ijms-1173341), entitled “The neutrophil secretome as a crucial link between inflammation and thrombosis”, Authors, Blanch-Ruiz et al, reviewed the studies on role of neutrophil-derived microvesicles (NMV) and neutrophil extracellular traps (NET) in inflammation and thrombosis through interaction with endothelial cells and platelets. This topic is important, because this mechanism could be related with cardiovascular diseases, such as myocardial infarction, stoke, and other cardiovascular events. However, there are several major concerns, which are listed in the following paragraphs:

  1. The font of words in the figure is too small. Please update the figures.
  2. In Page 6, lines 220-222, authors mentioned that macrovesicles in healthy individuals are thought to exert an anticoagulants unction by promoting the generation of low amounts of thrombin. Thrombin is a coagulation factor (F-IIa), which could induce coagulation and thrombosis, instead of anticoagulation. Why thrombin activate protein C only, not coagulation cascade, under this condition. Please address this issue.
  3. In page 5, lines 176-177, NMV activates both intrinsic and extrinsic coagulation pathways. However, the figure only shows extrinsic coagulation pathway.
  4. What are the mechanisms or risk factors which trigger NMV and NET formation, leading to cardiovascular events, such as stroke, myocardial infarction? Please discuss how neutrophils are activated in the patients with these diseases.

Author Response

Referee 3

In this manuscript (ID: ijms-1173341), entitled “The neutrophil secretome as a crucial link between inflammation and thrombosis”, Authors, Blanch-Ruiz et al, reviewed the studies on role of neutrophil-derived microvesicles (NMV) and neutrophil extracellular traps (NET) in inflammation and thrombosis through interaction with endothelial cells and platelets. This topic is important, because this mechanism could be related with cardiovascular diseases, such as myocardial infarction, stoke, and other cardiovascular events”.

We fully appreciate the comments of reviewer 3 and thank her/him for considering that “This topic is important, because this mechanism could be related with cardiovascular diseases, such as myocardial infarction, stoke, and other cardiovascular events”. The manuscript has been modified according to the referee’s suggestions, which we believe have helped to improve our work. The specific responses to the comments of the referee are as follows:

Point 1. The font of words in the figure is too small. Please update the figures.

Response 1. According to referee’s suggestion, figures have been updated and the font size of words in the figures has been increased (Page 3, line 102, figure 1; Page 5, line 202, figure 2; page 9, line 326, figure 3).

Point 2. In Page 6, lines 220-222, authors mentioned that macrovesicles in healthy individuals are thought to exert an anticoagulants unction by promoting the generation of low amounts of thrombin. Thrombin is a coagulation factor (F-IIa), which could induce coagulation and thrombosis, instead of anticoagulation. Why thrombin activate protein C only, not coagulation cascade, under this condition. Please address this issue.

Response 2. The point of the referee is valid as thrombin is a coagulation factor which can induce coagulation and thrombosis. Berckmans and colleagues (Berckman et al. Thromb Haemost 2001; 85: 639-646) indicate that microparticles present in circulation from healthy individuals have an anticoagulant function by promoting the generation of low amounts of thrombin and they point to another fundamental principle of coagulation; that low concentrations of thrombin are anti-coagulant by virtue of their capacity to generate activated protein C (Harker et al. J Clin Invest 1993; 92: 2003-2012). As it may be contradictory, this concept has been removed from the review.

Point 3. In page 5, lines 176-177, NMV activates both intrinsic and extrinsic coagulation pathways. However, the figure only shows extrinsic coagulation pathway.

Response 3. Figure 2 now shows both extrinsic and intrinsic coagulation pathways (Page 5, line 202, figure 2).

Point 4. What are the mechanisms or risk factors which trigger NMV and NET formation, leading to cardiovascular events, such as stroke, myocardial infarction? Please discuss how neutrophils are activated in the patients with these diseases.

Response 4. Following referee’s suggestion, the possible mechanisms by which neutrophils are activated and trigger NETs formation in patients with cardiovascular diseases, and particularly with myocardial infarction, have been discussed in “neutrophil extracellular traps” section (Page 9, line 342-346).

Round 2

Reviewer 2 Report

No further comments. The reviewers have addressed all my comments